# A Pragmatic Machine Learning Approach to Quantify Tumor-Infiltrating Lymphocytes in Whole Slide Images

**DOI:** 10.3390/cancers14122974

**Published:** 2022-06-16

**Authors:** Nikita Shvetsov, Morten Grønnesby, Edvard Pedersen, Kajsa Møllersen, Lill-Tove Rasmussen Busund, Ruth Schwienbacher, Lars Ailo Bongo, Thomas Karsten Kilvaer

**Affiliations:** 1Department of Computer Science, UiT The Arctic University of Norway, N-9038 Tromsø, Norway; nikita.shvetsov@uit.no (N.S.); edvard.pedersen@uit.no (E.P.); lars.ailo.bongo@uit.no (L.A.B.); 2Department of Medical Biology, UiT The Arctic University of Norway, N-9038 Tromsø, Norway; morten.gronnesby@uit.no (M.G.); lill.tove.rasmussen.busund@uit.no (L.-T.R.B.); ruth.schwienbacher@unn.no (R.S.); 3Department of Community Medicine, UiT The Arctic University of Norway, N-9038 Tromsø, Norway; kajsa.mollersen@uit.no; 4Department of Clinical Pathology, University Hospital of North Norway, N-9038 Tromsø, Norway; 5Department of Oncology, University Hospital of North Norway, N-9038 Tromsø, Norway; 6Department of Clinical Medicine, UiT The Arctic University of Norway, N-9038 Tromsø, Norway

**Keywords:** NSCLC, deep learning, digital pathology, tumor-infiltrating lymphocytes

## Abstract

**Simple Summary:**

Tumor tissues sampled from patients contain prognostic and predictive information beyond what is currently being used in clinical practice. Large-scale digitization enables new ways of exploiting this information. The most promising analysis pipelines include deep learning/artificial intelligence (AI). However, to ensure success, AI often requires a time-consuming curation of data. In our approach, we repurposed AI pipelines and training data for cell segmentation and classification to identify tissue-infiltrating lymphocytes (TILs) in lung cancer tissue. We showed that our approach is able to identify TILs and provide prognostic information in an unseen dataset from lung cancer patients. Our methods can be adapted in myriad ways and may help pave the way for the large-scale deployment of digital pathology.

**Abstract:**

Increased levels of tumor-infiltrating lymphocytes (TILs) indicate favorable outcomes in many types of cancer. The manual quantification of immune cells is inaccurate and time-consuming for pathologists. Our aim is to leverage a computational solution to automatically quantify TILs in standard diagnostic hematoxylin and eosin-stained sections (H&E slides) from lung cancer patients. Our approach is to transfer an open-source machine learning method for the segmentation and classification of nuclei in H&E slides trained on public data to TIL quantification without manual labeling of the data. Our results show that the resulting TIL quantification correlates to the patient prognosis and compares favorably to the current state-of-the-art method for immune cell detection in non-small cell lung cancer (current standard CD8 cells in DAB-stained TMAs HR 0.34, 95% CI 0.17–0.68 vs. TILs in HE WSIs: HoVer-Net PanNuke Aug Model HR 0.30, 95% CI 0.15–0.60 and HoVer-Net MoNuSAC Aug model HR 0.27, 95% CI 0.14–0.53). Our approach bridges the gap between machine learning research, translational clinical research and clinical implementation. However, further validation is warranted before implementation in a clinical setting.

## 1. Introduction

Increasing the availability of digital pathology opens new possibilities. Digital whole slide images (WSIs) stored on servers are easily retrieved for review by pathologists. Viewing serial WSIs from the same patient side-by-side enables the pathologist to assess the morphology and protein expression at the same time, as opposed to swapping the slides back-and-forth when viewing with a microscope. As a bonus, the field of view is vastly increased compared to the ~1000-μm circle offered at 400× in a microscope. Further, residents in pathology may demark areas of interest and pass these to consultant pathologists for feedback and learning.

Although digital pathology will optimize the current work flows, computational pathology is by many believed to be one of its most significant advantages. A standard three-channel WSI is approximately 100,000 × 100,000 pixels. This represents an enormous amount of information that may be used to identify, and subsequently quantify, macro- (patches of cartilage and bone, islets of cancer or vessels); intermediate- (different types of cells) and micro-structures (different cellular components). Whilst some of this information is utilized in classical light microscopy, important information is likely discarded. Moreover, image features, abstract to the human mind, may be extracted and incorporated as biomarkers into existing diagnostic pipelines, expanding the prognostic and predictive toolkit. An additional benefit may be a reduction of intra- and interobserver variations [1]—a known diagnostic challenge that is especially pronounced if the problem is complex and/or open for personal opinions.

A hematoxylin & eosin (H&E)-stained tissue slide is the principal method used to evaluate tissue morphology, while immunohistochemistry (IHC), with chromogens or fluorophores bound to an antibody, is used to visualize specific protein expressions. Classically, the focus has been on the morphology and protein expression of cancer cells. However, numerous studies have highlighted the interactions between cancer cells and their surroundings. This interplay, popularly termed the tumor microenvironment, impacts patients’ prognoses and is likely important when deciding treatment strategies. Immune cells and, especially, tumor-infiltrating lymphocytes (TILs) are among the most important cells in the tumor microenvironment and will, in many cases, directly influence cancer development. As a result, different variations of TILs have been suggested as prognostic and/or predictive biomarkers in various types of cancers, including colorectal cancer, breast cancer, melanoma and non-small cell lung cancer (NSCLC) [2,3,4,5]. Interestingly, the strategies for TIL identification vary between cancer types. In colorectal cancer (CRC), IHC is used to visualize CD3+ and CD8+ TILs [4], while conventional H&E is used for TIL identification in breast cancer and melanoma [2,3]. Moreover, TIL quantification ranges from fully discrete grouping via semi-quantitative to absolute count-based methods. For other cancers, such as NSCLC, the preferred method of TIL identification and quantification is yet to be determined [2,5,6]. While IHC-based methods provide information about the cell type and function, they add complexity, time spent and cost. Hence, using conventional H&E for TIL identification is tempting, especially if identification can be conducted without time-consuming manual counting.

Computational pathology presents an opportunity for automatic TIL identification in H&E WSIs. The methods range from simple rule-based systems, which rely on easily understandable hand-crafted features such as shape, size and color, to deep learning (DL) approaches, which calculate and combine image features nonlinearly. Given proper implementation, DL will usually outperform the sensitivity/specificity of simpler approaches at the cost of increased complexity, calculation time and loss of interpretability. Unfortunately, DL requires thousands of annotated images to reach its full potential, and the procurement of annotated training datasets may be challenging due to the required pathologist effort. Fortunately, several recent projects provided datasets and introduced highly accurate DL models for cell segmentation and classification in H&E images [7,8,9,10,11,12,13]. However, to our knowledge, none of these approaches have been adapted to TIL quantification in a clinical setting.

Herein, we present our approach and cloud-based system for the quick evaluation and deployment of machine learning models in a clinical pathology setting using our real-world dataset from patients with NSCLC. To investigate the clinical usefulness of DL for TIL quantification, we:Implemented the HoVer-Net [7] algorithm and compared our results to the original paper and the work of Gamper et al. [10].Tested how HoVer-Net performs on unseen data, which will be the case in a clinical setting.Evaluated an additional augmentation approach with the aim of improving the performance on unseen data [14].Compared the prognostic performance of TILs identified by HoVer-Net to the standard state-of-the-art approach (CD8 staining).Assessed the immune cell annotations produced by six HoVer-Net models trained on three datasets for our unlabeled dataset comprising H&E-stained images of NSCLC.Created a system for deploying machine learning models in a cloud to support the implementation of computational pathology and to quickly evaluate promising methods in a clinical setting.

## 2. Materials and Methods

We present our approach and cloud-based system for transferring open-source machine learning models trained on public datasets to solve new problems in H&E WSI annotation. As a proof-of-concept, we customized the HoVer-Net algorithm [7] to detect and quantify TILs in tissue samples from patients diagnosed with NSCLC. However, the developed pipeline is easily adapted to any object detection task in WSIs. To validate the customized model performance on our data without annotating hundreds of thousands of cells, we used a visual inspection of the inference results and confirmed their clinical utility in 1189 1000 × 1000-μm patches at a resolution of 0.2428 μm/px from 87 NSCLC patients.

## 3. Datasets

### 3.1. Public Datasets to Train and Test a Machine Learning Model

In the past decade, several excellent annotated datasets for cell segmentation and classification were made publicly available, as summarized in Table 1. The CoNSeP, PanNuke and MoNuSAC datasets were, at the time of submission, the only publicly available datasets comprising both the segmentation masks and class labels necessary to train DL models for simultaneous cell segmentation and classification. The dataset characteristics are summarized in Table 1, and a simplified version of how patches are generated and processed during the training procedure is illustrated in Figure 1.

### 3.2. UiT-TILs Dataset Used to Clinically Validate TIL Classifications

To validate the clinical relevance of TIL detection, we compiled the UiT-TILs dataset comprising 1189 image patches from 87 NSCLC patients with matched clinical data. The UiT-TILs dataset is a subset of the cohort presented by Rakaee et al. in 2018 [15]. The patches were generated in QuPath [16] using thresholding to detect the tissue area, followed by the generation of 4019 × 4019 px (1000 × 1000-μm) patches and the manual selection of up to 15 consecutive patches from the tumor border. Due to differences in the tumor size, tissue quality and the absolute length of the tumor border present in each WSI, the target number of 15 patches from each patient could not be met in all instances (median number = 15, range 3–16). The patient’s clinical data are summarized in Appendix A.

### 3.3. Augmentation

Our augmentation strategy includes the use of a heuristic data augmentation scheme thought to boost the classification performance [17]. First, we augment the two training datasets using the augmentation policies provided in the HoVer-Net source code [7]. These include affine transformations (flip, rotation and scaling), Gaussian noise, Gaussian blur, median blur and color augmentations (brightness, saturation and contrast). Second, we introduce a linear augmentation policy proposed by Balkenhol et al. [14] for better generalizability and overall improvement of the trained models. The image is (1) converted from the RGB to the HED color space using the scikit image implementation of color deconvolution, according to the Ruifrok and Johnston algorithm [18]; (2) each channel is transformed using a linear function that stochastically picks coefficients from a predefined range, and (3) transformed channels are combined and converted back into the RGB space.

### 3.4. Replicated Training Algorithm

We repurpose the training pipeline from the HoVer-Net source code. In brief, we preprocess the data and extract sub-patches from the training dataset. For the CoNSeP, MoNuSAC and PanNuke datasets, the input patch sizes are 270 × 270 px, 256 × 256 px and 256 × 256 px, respectively. For each input patch, the RGB channels are normalized between 0 and 1. The datasets are augmented via predefined policies and handled by the generator during training. Figure 1 is a simplistic depiction of preprocessing and training. The training procedure is performed in two phases—training decoders and freezing encoders and fine-tuning of the full model. The training is initialized on weights pretrained on ResNet50 and optimized using Adam optimization while decreasing the learning rate after 25 epochs for each phase. The models are trained for a total of 100 epochs. The ConSeP dataset used by Graham et al. is split into training and test subsets with 27 and 14 image patches, respectively. Since no validation set is provided for the CoNSeP dataset, we use the training and test sets to train and validate the CoNSeP models, and the test set is used again to evaluate the model performance. The MoNuSAC dataset provides training and test sets, and when training our models, we split the training set 80/20 into training and validation sets, respectively. The PanNuke dataset provides separate sets for the training, validation and testing models. For additional details about the training parameters, we refer to the original paper and their open-source code [7,10,11].

### 3.5. A System for Training, Tuning and Deploying Machine Learning Models

To easily tune and evaluate machine learning models, we need an environment that has the computing resource to keep the training time short, provide easy deployment, keep track of the experimental configuration and ensure that our experiments are reproducible. In addition, we want to make the code portable. This will enable us to deploy the containers on a commercial cloud with GPU support to gain access to the needed computing resources for fast training or on a secure platform to train on sensitive data that cannot be moved to a commercial cloud.

To solve the above challenges, we containerize the code so that we can easily deploy it on a cloud platform or our in-house infrastructure. We use Docker [19] containers and a Conda [20] environment for dependency handling. We use Kubernetes [21] for container management. To ensure the experiment reproducibility and to keep track of the tuning parameters, we use Pachyderm [22], since it enables data and model provenance and parallelized batch processing.

We deploy our customized HoVer-Net model using our approach summarized in Figure 2. To make it easier to test different training data, augmentation and parameters, we use configuration profiles implemented as YAML files generated in the process and stored in the code repository. We split data wrangling and training from inferences to make it easier to deploy these on a new platform. To reduce the execution time, we use Tensorpack with Tensorflow 1.12 to infer and post process patches in parallel.

We use a docker container on a local server with a RTX Titan GPU for development and debugging. We train and tune the models on the Azure cloud, using AKS (Azure Kubernetes Service) with GPU nodes with the Pachyderm framework. As a node pool, we use 1 Standard NC6 (6 cores, 56 GB RAM) node with 1× K80 GPU accelerator.

### 3.6. Interactive Annotation of WSIs

To demonstrate and visually evaluate the TIL classifications and quantification, we developed the hdl-histology system (Figure 3). It is an interactive web application for exploring and annotating gigabyte-sized WSIs. Our web server implements the URL endpoints required to request and transfer WSI tiles with interactive response times. It uses the Flask framework to implement a RESTful API. We use Gunicorn (Green Unicorn) [23] to handle concurrent requests. We use OpenSeadragon (OSD) to implement a tiled map service [24]. To achieve an interactive performance for high-resolution WSIs, we export the slides from their native format (.vms, .mrxs and .tiff, etc.) using *vips dzsave* and generate Microsoft’s DeepZoom Image (.dzi) tiles as a pyramidal folder hierarchy of .jpegs on a disk. We provide classification as a service for interactive patch annotation. We use Tensorflow Serving as a containerized server for the trained models. This service exports a REST API. The input is the patch to process and the name of the model to use. The service returns a class probability map that can be used to create a visual mask for the cells in the patch. The REST API is used by the webserver that overlays the returned pixel matrix of the annotated regions.

The web server runs in a docker container on the Azure App Service. The app service pulls the docker image from our Azure Container Registry. The images are stored in a blob container, which is path mapped to our app service. To deploy updates to the system, we use continuous integration with azure pipelines. The deployment pipeline triggers push events of an azure branch in the git repository using GitHub Webhooks (https://docs.github.com/en/developers/webhooks-and-events/about-webhooks, accessed on 1 August 2019). The build pipeline pulls the latest change from the branch and builds a docker container from the repos Dockerfile and pushes the container image into our Azure Container Registry (Figure 2).

### 3.7. Validation of TIL Quantification on Unlabeled Data

To pragmatically evaluate the quantification of TILs without the large effort required for manual labeling, we apply three methods.

First, to verify that our modifications to HoVer-Net do not change the results, we reproduce the results described in the original HoVer-Net and PanNuke papers using the CoNSeP and PanNuke datasets (Appendix A) [7,10]. Further, to investigate whether an additional augmentation leads to more robust and generalizable models, we use the CoNSeP, MoNuSAC and PanNuke datasets to train separate models with and without additional augmentations.

For comparison to the original HoVer-Net paper, we evaluate the segmentation and classification tasks using the metrics provided in the HoVer-Net source code [7]. The full description of the metrics used in the replication study is available in the Appendix A.

For all other comparisons, we use the conventional and well-established metrics to compare and benchmark our models. Since the models encompass both segmentation and classification, we compare the segmentation and classification separately and combined. The definitions of the metrics we used to evaluate the segmentation and classification separately are available in the Appendix A. The integrated segmentation/classification metrics include:(1)$$Accuracy_{dc}^{t}=\frac{TP_{dc}^{t}+TN_{dc}^{t}}{TP_{dc}^{t}+TN_{dc}^{t}+FP_{dc}^{t}+FN_{dc}^{t}}$$
(2)$$Precision_{dc}^{t}=\frac{TP_{dc}^{t}}{TP_{dc}^{t}+FP_{dc}^{t}}$$
(3)$$Recall_{dc}^{t}=\frac{TP_{dc}^{t}}{TP_{dc}^{t}+FN_{dc}^{t}}$$
(4)$$F1_{c}^{t}=\frac{TP_{dc}^{t}}{TP_{dc}^{t}+0.5\left(FPdc^{t}+FN_{dc}^{t}\right)}$$
where $TP_{dc}^{t}$- detected cells with the GT label *t*, classified as *t*, $TN_{dc}^{t}$- detected or falsely detected cells with a GT label other than *t*, classified as other than *t*, $FP_{dc}^{t}$- detected or falsely detected cells with a GT label other than *t*, classified as *t*, and $FN_{dc}^{t}$- detected cells with the GT label *t*, classified as other than *t* and all cells with the GT label *t* not detected for each class *t*.

As t, we use inflammatory and cancer cells. To compare our augmentation approach with the original approach we applied the trained models to their respective test datasets and used cross-inference with the test data from the other dataset. For an evaluation of the cross-inference results, we had to count both normal epithelial and neoplastic cells as cancer cells, due to the fact that the CoNSeP and MoNuSAC datasets do not differentiate between normal and neoplastic epithelial cells.

Second, to demonstrate that our methods may be adapted for use in a clinical setting, we apply our models on all patches in the UiT-TILs dataset and calculate the median number of cells classified as inflammatory/TILs for each patient. The median number of identified immune cells will then be correlated to the number of immune cells identified using digital pathology and specific immunohistochemistry on tissue microarrays and semi-quantitatively scored TILs previously evaluated in the same patient cohort [15,25]. To assess the impact of TILs identified using our deep learning approach, the patients are stratified into high- and low-TIL groups, and their outcomes are compared using the Kaplan–Meier method and the log–rank test with disease-specific survival as the endpoint.

Third, to demonstrate that our methods are meaningful in a clinical pathology setting, two experienced pathologists (LTRB and RS) visually inspected the segmentation and classification results for the TILs and cancer cells in 20 randomly sampled patches from the provided dataset. During the inspection, they estimated the precision and recall for the TILs and cancer cells using 10% increments for each measurement. We calculate the F1 scores based on the estimated precision and recall for each patch.

## 4. Results

### 4.1. Model Performance

We trained a total of three pairs of HoVer-Net models with and without additional augmentations using training data from: (1) CoNSeP, (2) MoNuSAC and (3) PanNuke. Each model’s performance was evaluated on its respective test set and on the test set of the other models. Additional information on the training and inference times are available in the Appendix A.

First, we verified that our replication of the original HoVer-Net models, trained on the CoNSeP and PanNuke datasets, provided comparable performances to the results presented by Graham et al. [7] and Gamper et al. [10]. The results are presented in Appendix A and show that our implementations performed within the expected range compared to the original studies’ results. We believe the small differences are due to missing details in the original studies (for ConSeP, the partitioning and preprocessing is not clearly described; for PanNuke, the source code and model hyperparameters are not available).

Second, to get an estimation on real-world model performance, we evaluated our original HoVer-net models trained on CoNSeP, MoNuSAC and PanNuke data on the corresponding test sets for each dataset. As expected, all models suffered a drop in performance when used for inference on a test set generated from another data source. This effect was most prominent for the model trained on ConSeP data, whose performance dropped significantly when tested on MoNuSAC and PanNuke data (Table 2 and Appendix A, columns BI and CI vs. AII). The performance drop was especially prominent during classification and was consistent for both cancer and inflammatory cells. A similar trend was observed for the model trained on MoNuSAC data (Table 2 and Appendix A, columns CV vs. AV and BV). The PanNuke model tested on ConSeP and MoNuSAC data (Table 2 and Appendix A, column BIII vs. AIII and CIII) exhibited less variable results on different data.

Third, we evaluate the effect of our augmentation policies on the cross-inference performances, since better augmentation often improves model transferability. However, our results do not indicate a significant performance improvement of the CoNSeP model (Table 2, columns B I vs. B II and CI vs. CII), the MoNuSAC model (Table 2, columns AV vs. AVI and BV vs. BVI) or the PanNuke model (Table 2, columns AIII vs. AIV and CIII vs. CIV).

Finally, we compare the augmented models (Table 2). The CoNSeP and MoNuSAC models perform best on their own data and experience performance drops when used on unseen data. Interestingly, the PanNuke model performs better on MoNuSAC data and compares favorably with models trained on the CoNSeP and MoNuSAC datasets when tested on their data. Consequently, we believe the PanNuke test data results are the most representative of the expected performance on lung tumor tissue in the UiT-TIL data, since PanNuke includes patches with this tissue type. We therefore expect PanNuke to perform better on UiT-TILs.

### 4.2. Correlation of TIL Quantification with Cancer Survival Rate

We compared the number of TILs identified with the PanNuke Aug model, with the number of TILs identified using our simple rule-based approach (*Helm*, described in the Appendix A). HoVer-Net models trained on the CoNSeP, MoNuSAC and PanNuke datasets with and without additional augmentations and different subsets of T-lymphocytes in TMAs previously investigated in the same patients (Kilvaer et al. 2020) (Appendix A). TILs identified using the rule-based approach (R 0.44) and with the CoNSeP (Original R 0.54; Aug R 0.92), MoNuSAC (Original R 0.89; Aug R 0.90) and PanNuke models (Original R 0.94) were moderately-to-strongly correlated with the TILs identified with the PanNuke Aug model. Moderate correlations were also observed with the number of TILs identified by the pan T-lymphocyte marker CD3 and the cytotoxic T-cell marker CD8 (R-values of 0.54 and 0.39, respectively) in TMAs from the same patients.

We evaluated the clinical impact of the TILs identified using our approach in the UiT-TIL cohort. The results are summarized in Figure 4 and in Appendix A. In brief, we show that TILs, identified by either the original and augmented MoNuSAC and PanNuke models or the augmented CoNSeP model, can be used to identify TILs as prognostic factors in NSCLC (Appendix A and Figure 4E–G). Specifically, patients with an above median number of TILs present with superior survival and compared favorably with state-of-the-art CD8 lymphocyte detection in the same patients (Appendix A and Figure 4C). This shows that a deep learning model could be used instead of CD8 staining to quantify TILs.

### 4.3. Manual Validation of Classification Results

Two pathologists, LTRB and RS, manually validated the performances of three original and three augmented HoVer-Net models in a real-world dataset comprising 20 patches. The results for all the patches are provided in Appendix A. For cancer cells, the recall was >90% for all models, while the precision was variable. For inflammatory cells, the recall was variable, while the precision was >90% for all models. Figure 5 exemplifies the best, worst and largest range for the classification of the individual patches according to our visual validation. Both pathologists reported that, for cancer cells, precision seemed to be directly correlated to the number of cancer cells, while, for immune cells, recall seemed to be inversely correlated to the number of immune cells.

## 5. Discussion

### 5.1. Summary of the Results

In this paper, we presented our adaptation of HoVer-Net. Specifically, we implemented the original approach as presented by Graham et al. and retrained the model on an expanded dataset published by Gamper et al. [7,10]. To increase the model accuracy and prevent overfitting, we introduced additional augmentations and compared models trained with and without these. Interestingly, our additional augmentations did not lead to a measurable increase in the model performance (Table 2 and Appendix A). Moreover, we published UiT-TILs, a novel retrospective dataset consisting of patient information and tumor images sampled from 87 patients with NSCLC. We used this dataset to explore the possible clinical impact of our approach and compared it with the current state-of-the-art methods for TIL evaluation in NSCLC patients. Our results suggest that the prognostic impact of TIL identification in H&E-stained sections with the current DL approach is comparable, and possibly superior to, the chromogenic assays (standard IHC-stained CD8 cells in TMAs HR 0.34, 95% CI 0.17–0.68 vs. TILs in HE WSIs: HoVer-Net PanNuke Aug Model HR 0.30, 95% CI 0.15–0.60, and HoVer-Net MoNuSAC Aug model HR 0.27, 95% CI 0.14–0.53, Figure 4 and Appendix A). Finally, to provide an environment for easy and fast prototyping and deployment for experiments and production, we built a cloud-based WSI viewer to support our back end. Our system is scalable, and both the front end and back end can function separately.

### 5.2. Related Works and Lessons Learned

The number of potential use cases for digital pathology is increasing. However, while the basic research is making strong headway, the use of digital pathology in routine clinical pathology is currently not aligned with the expectations for most laboratories. In a recent paper, Hanna et al. reviewed the process of adopting digital pathology in a clinical setting and shared from their process at the Department of Pathology, Memorial Sloan Kettering Cancer Center [26]. They concluded that, to increase the number of pathologists willing to adopt digital pathology, the technology has to offer benefits that impact the pathologist’s daily work. While conventional image analyses have proven suitable for some tasks, such as counting the number of cancer cells staining positive for Ki67 in breast cancer [27], they are prone to failure, given the increasing complexity. We, among many others, believe that DL will deliver the tools necessary to irreversibly pave the way for digital pathology. Despite this optimism, one of the main arguments against DL in medicine in general, and in cancer care in particular, is its inherent black box nature. As an example, Skrede et al. published DoMore-v1-CRC, a prognostic model in colorectal cancer that outperformed the prognostic impact of the TNM staging system for CRC patients in retrospective data [28]. Moreover, their model may help select patients that will benefit from adjuvant chemotherapy. However, whether their lack of explanation will impede its implementation remains to be seen. We chose to focus on cell segmentation and classification in general and on TILs in particular. Distinct from models that derive patients’ prognoses from abstract features from imaging data alone, the success of our model can be evaluated directly based on its output (Figure 5). Interestingly, the prognostic impact derived from TILs identified on NSCLC H&E WSIs by the current state-of-the-art segmentation and classification models are equal or superior to the IHC-based methods. This latter finding was corroborated by our manual validation, where the segmentation and classification of the TILs were consistent through the majority of the images. However, we observed a significant difference in the model performances for an image with a darker background and overall worse performances in images where TILs were scarce. These latter points suggest that our models are overfitting on the dataset. To summarize, our present effort illustrates that it is possible to repurpose existing DL algorithms for use on real patient samples with understandable output translating into clinical information that may be used to make informed treatment-related decisions for patients.

In an attempt to further generalize our models, we adapted a linear augmentation policy proposed by Balkehol et al. to the HoVer-Net training pipeline [14]. For models trained on the PanNuke and MoNuSAC datasets, we did not observe the expected benefits of the additional augmentations (Appendix A). For models trained on the CoNSeP dataset, there was a tendency towards better performances when additional augmentations were used. Currently, there is no reference augmentation policy for use on histopathology datasets. Our experience suggests that smaller datasets may benefit the most from additional augmentation. However, there are several steps, including, but not limited to, tissue handling, staining, scanning, normalization and additional augmentations fitted before or during training, that may impact the model performance. Exploring these is beyond the scope of the current paper, but our use of MLOps best practices should make it easy to evaluate different normalization techniques and augmentation policies. Based on the reasonable performances of the CoNSeP models on CRC tissues and on the CoNSeP dataset, another feasible approach is to generate specific datasets for each problem. This latter strategy may make models less complex, because the variability will be reduced. However, we do not believe that task-specific datasets will contribute to more accurate TIL detection, as TILs, contrary to cancer cells, are morphologically equal across all cancers.

### 5.3. Future Work

As a general direction for future works, we can outline the following points: prune and optimize models to run faster with less computational power; improve web server applications to reduce response times; integrate with widely used tools such as ImageJ/Fiji, Cell Profiler, QuPath and others; add additional features useful as decision support for pathologists and test acquired models on new datasets to validate the model performance on other tissue types. It is also pertinent to explore new methods for dataset generation that incorporate all the steps from tissue handling to staining to segmentation, auto-labeling and manual and/or automatic feedback.

For the specific use case of TIL segmentation and classification, future studies should address localization and the possible varying distributions of TILs in a tumor. Further, tile sampling may reduce the amount of tissue needed to be assessed and thereby compute resources. In addition, incorporating additional elements, such as the class of surrounding cells and information of the tissue structure, may further improve the performance of the classification step.

### 5.4. Limitations

In our composed dataset, we manually extracted patches to test our models and hypotheses. In future works, we plan to eliminate that flaw and automatically extract the region of interest in the WSI.

In this work, our aim was to provide an interactive graphical user interface for pathologists able to test and evaluate different machine learning models for a WSI analysis. The response time is minutes for average size patches and hours for WSIs. It can be improved by using a more powerful GPU or multiple GPUs. However, we also want to make the service accessible to pathologists by using cloud services. The cloud service cost may be significant, especially if the service is running continuously. Model optimization may reduce the cost and improve the response times by using, for example, hardware accelerators or pruning techniques. It may also be more cost-efficient to deploy the systems on an on-premise server with GPUs. Currently, our tool is designed for hypothesis generation, testing and education. In a clinical implementation, we need to improve the response times, reduce the costs, make it easier and more convenient to use, integrate it with the existing clinical ecosystems, test it in clinical scenarios and obtain permission from regulatory bodies for its use as a medical device.

## 6. Conclusions

In conclusion, we used already published state-of-the-art DL models for the segmentation and classification of cells and adapted and retrained these for use on new data. We showed that TIL identification using DL models on H&E WSIs is possible and, further, that the clinical results obtained are comparable to, and possibly supersede, the prognostic impact of conventional IHC-based methods for TIL identification in tissues from lung cancer patients. Moreover, our approach, utilizing published datasets, significantly limits the need for creating additional costly labeled datasets. Distributed deployment and deployment as a service allow for easier and faster model deployment, potentially facilitating the use of DL models in digital pathology. However, before clinical implementation, the efficacy of our methods needs to be further validated in independent datasets, and the system must be tightly integrated into the clinical work flow to ensure adaptation to the daily diagnostics.

## Figures and Tables

**Figure 1 cancers-14-02974-f001:**
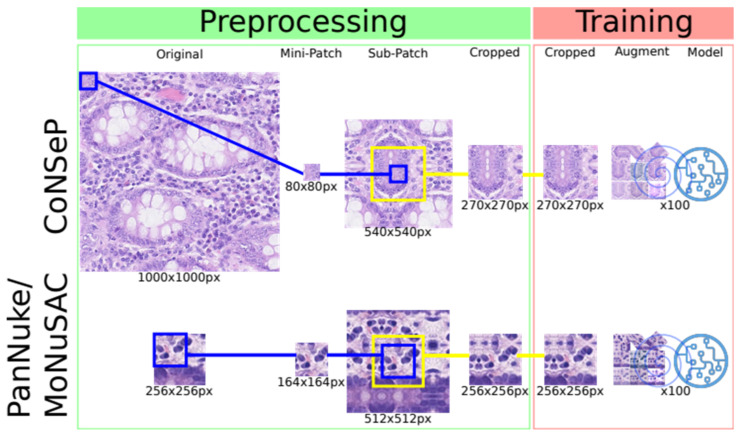
For each dataset, the original patches are divided into nonoverlapping (CoNSeP) and overlapping (PanNuke/MoNuSAC) mini-patches. A sub-patch is generated by adding padding using image information from adjacent tissue in the original image patch and/or by mirroring in edge cases. The central part of the sub-patch is cropped and used in the training procedure. The output of the network (segmented/classified cells) is equal to the mini-patch. Each model is subsequently trained for 100 epochs.

**Figure 2 cancers-14-02974-f002:**
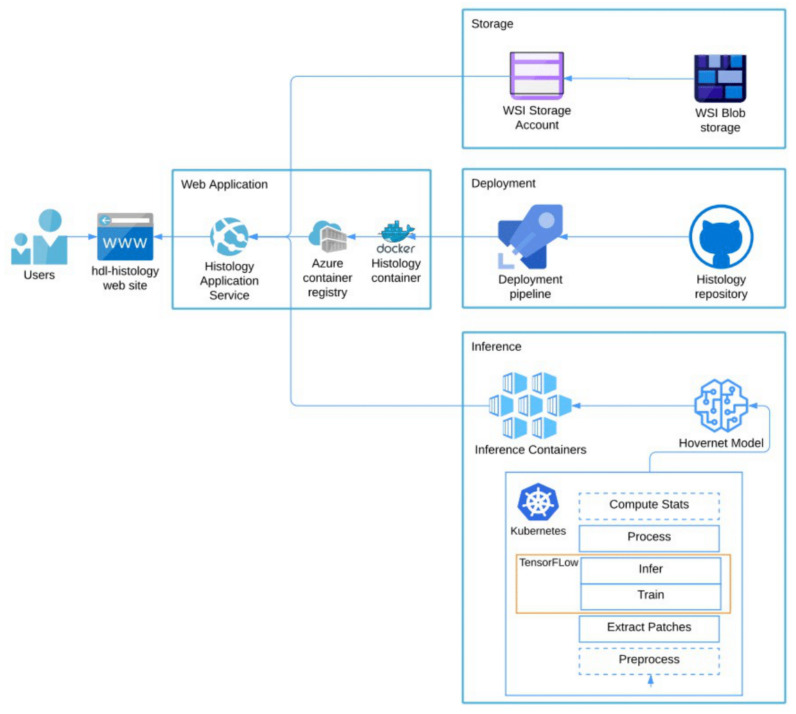
Overview of cloud deployment of the web application. WSIs are uploaded into Azure Blob storage and mapped as a file system in the web application. The web application is built by an Azure Pipeline from a docker file contained in the histology git repository. We use the Kubernetes platform in Azure to train and validate our implementation of HoVer-Net. The exported models are implemented as inference containers in the histology application service and served using Tensorflow Serving.

**Figure 3 cancers-14-02974-f003:**
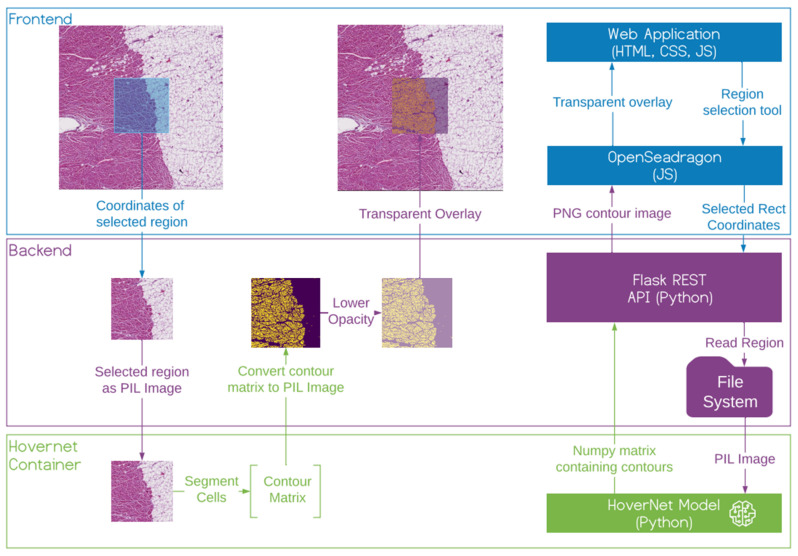
Overview of the slide viewer front end, back end and inference container. Selected regions of the front end are cropped out at the back end and transferred to the inference container. The inference container returns a contour matrix that is converted into a contour image and overlaid on top of the front end WSI.

**Figure 4 cancers-14-02974-f004:**
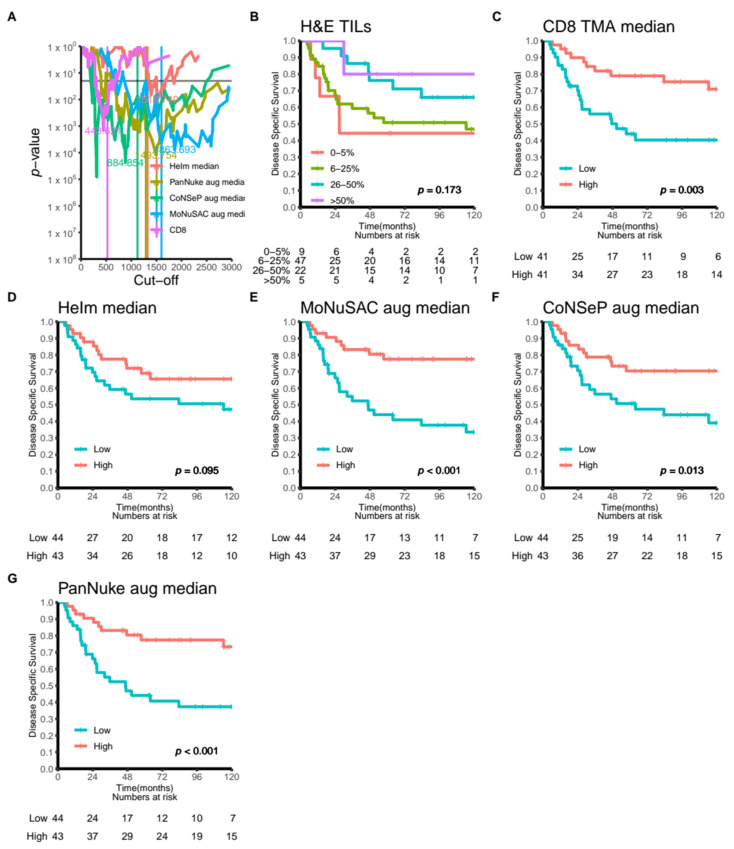
(**A**) All possible dichotomized cut-offs for the TILs identified by CD8 IHC in TMAs or using the HeIm, PanNuke or CoNSeP models plotted against *p*-values indicating the significance of the DSS for all included patients (*n* = 87). (**B**–**G**) Disease-specific survival curves for high- and low-TIL scores for (**B**) the semi-quantitative model proposed by Rakaee et al. 2018. (**C**) The CD8 model proposed by Kilvaer et al. 2020. (**D**–**F**) High and low numbers of TILs identified using the baseline rule-based HeIm algorithm (described in the Appendix A) or DL models trained on the CoNSeP, MoNuSAC and PanNuke datasets, respectively.

**Figure 5 cancers-14-02974-f005:**
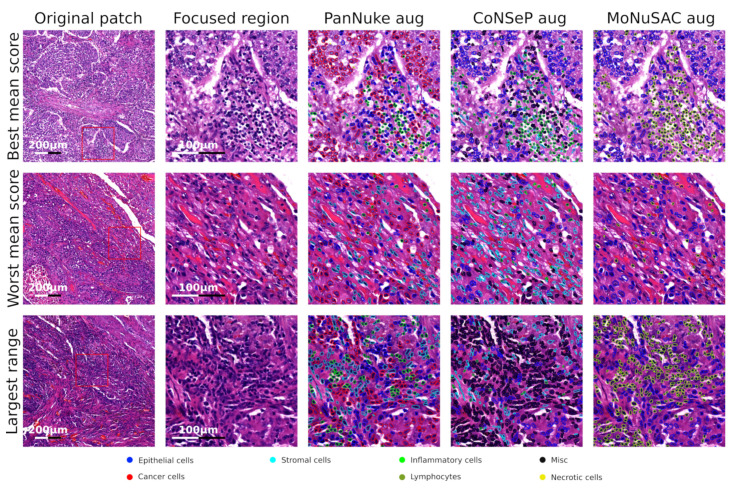
The top, middle and bottom rows represent the best, worst and largest range of F1 scores for the immune cell classifications from our manual validation, respectively. The first column is an overview of the entire image patch, while the second to fifth columns represent a focused region with and without detection/classification overlays.

**Table 1 cancers-14-02974-t001:** Characteristics of the datasets providing annotations and/or classifications publicly available for the training and validation of models for instant cell segmentation and classification.

Dataset	CoNSeP	PanNuke	MoNuSAC	CRCHisto *	TNBC	MoNuSeg	CPM-15	CPM-17
Author	Graham	Gamper	Verma	Sirinukunwattana	Naylor	Kumar	Vu	Vu
Year	2019	2020	2020	2019	2017	2017	2019	2019
Origin	UHCW	UHCW/TCGA	TCGA	UHCW	Curie Institute	TCGA	TCGA	TCGA
Tissue types	CRC	Various (19)	Various (4)	CRC	TNBC	Various (8)	Various (2)	Various (4)
Unique patients								
Number of patches	41	7901	294	100	50	30	15	64
Training	27 §	2722	209	NA	NA	16 #	NA	32
Validation	§	2656	NA	NA	NA	#	NA	NA
Testing	14	2523	85	NA	NA	14 #	NA	32
Patch size	1000 × 1000	256 × 256	Various	500 × 500	512 × 512	1024 × 1024	400 × 400 up to 600 × 1000	500 × 500 up to 600 × 600
Scanner(s)	Omnyx VL120	Various	Various	Omnyx VL120	Philips Ultra-Fast Scanner 1.6RA	Various	Various	Various
Magnification	40×	40×	40×	20×	40×	40×	40× and 20×	40× and 20×
Resolution	0.275 µm/px	Various	NA	0.55 µm/px	0.245 µm/px	NA	NA	NA
Annotation	NC	NC	NC	CoN	NC	NC	NC	NC
Cells	24,319	205,343	46,000	29,756	4022	21,623	2905	7570
Labeled cells	24,319	205,343	46,000	22,444	NA	NA	NA	NA
Cell types	7	5	4 (5)	4	NA	NA	NA	NA

Abbreviations: CoNSeP, CRCHisto, colorectal histology; PanNuke, MoNuSAC, TNBC, triple-negative breast cancer; MoNuSeg, CPM, NC, nucleus contour; UHCW, University Hospital; TCGA, The Cancer Genome Atlas; CoN, center of nucleus; NG, not given. * CRCHisto only provides center of the nucleus annotations. § Graham et al. 2019 suggested splitting the 27 training images into a training and validation set. However, the paper does not provide information on the split conducted in their published paper. # Kumar et al. 2017 suggested splitting into a training/validation set of 16 and 14 images. They also provided the split conducted in their published paper.

**Table 2 cancers-14-02974-t002:** A summary of the performance of four deep learning models trained using the CoNSeP (A I and II and B I and II) and the PanNuke (A III and IV and B III and IV) datasets using the original training pipeline as published by Graham et al. without (A I and III and B I and III) and with (A II and IV and B II and IV) enhanced augmentation. The best results for each parameter (1) within each dataset are in bold, and (2) models trained on another dataset are in italics. Separate comparisons of the segmentation and classification steps are provided in Appendix A.

Test Data	CoNSeP						PanNuke						MoNuSAC					
Model	CoNSeP		PanNuke		MoNuSAC		CoNSeP		PanNuke		MoNuSAC		CoNSeP		PanNuke		MoNuSAC	
Augmentation	HoVer	Aug	HoVer	Aug	HoVer	Aug	HoVer	Aug	HoVer	Aug	HoVer	Aug	HoVer	Aug	HoVer	Aug	HoVer	Aug
Numbering	AI	AII	AIII	AIV	AV	AVI	BI	BII	BIII	BIV	BV	BVI	CI	CII	CIII	CIV	CV	CVI
Integrated segmentation and classification																		
$Accuracy_{dc}^{inflammatory}$	0.78	**0.80**	0.71	0.71	*0.72*	0.55	0.73	** *0.76* **	0.71	0.72	0.47	0.46	0.59	0.70	0.74	** *0.75* **	0.73	0.73
$Precision_{dc}^{inflammatory}$	0.81	0.77	0.60	0.70	** *0.84* **	0.79	*0.59*	0.52	0.61	**0.68**	0.48	0.45	0.86	0.79	0.78	** *0.92* **	0.77	0.77
$Recall_{dc}^{inflammatory}$	0.58	0.66	0.79	0.74	** *0.83* **	0.60	0.28	0.36	**0.64**	0.61	0.53	*0.57*	0.26	0.49	0.78	*0.80*	0.86	** *0.87* **
$F1_{dc}^{inflammatory}$	0.68	0.71	0.68	0.72	** *0.83* **	0.68	0.38	0.42	0.63	**0.64**	*0.50*	*0.50*	0.40	0.60	0.78	** *0.86* **	0.81	0.82
$Accuracy_{dc}^{cancer}$	0.57	**0.59**	*0.54*	0.51	0.38	0.38	0.42	*0.45*	0.57	**0.58**	*0.45*	*0.45*	0.53	*0.63*	*0.63*	0.62	0.68	**0.70**
$Precision_{dc}^{cancer}$	0.59	0.61	0.61	0.62	0.68	** *0.72* **	0.53	0.50	0.64	0.63	0.70	** *0.72* **	*0.70*	0.62	0.58	0.57	0.71	**0.73**
$Recall_{dc}^{cancer}$	**0.74**	0.73	*0.68*	0.66	0.45	0.44	0.26	0.34	0.68	**0.73**	*0.51*	0.50	0.10	0.42	0.92	** *0.93* **	0.84	0.85
$F1_{dc}^{cancer}$	**0.66**	0.66	*0.64*	*0.64*	0.54	0.55	0.35	0.40	0.66	**0.68**	*0.59*	*0.59*	0.18	0.50	*0.71*	*0.71*	0.77	**0.79**

## Data Availability

Web server repository: https://github.com/uit-hdl/histology (accessed on 11 November 2021). Machine learning pipeline: https://github.com/uit-hdl/hovernet-pipeline (accessed on 18 March 2021) and convenience tools for easy setup: https://github.com/uit-hdl/hover_build (accessed on 25 March 2021). Publicly available datasets that we used to train our models: https://warwick.ac.uk/fac/cross_fac/tia/data/hovernet (ConSep, accessed on 1 October 2020), https://warwick.ac.uk/fac/cross_fac/tia/data/pannuke (PanNuke, accessed on 1 October 2020) and https://monusac-2020.grand-challenge.org/Data/ (MoNuSAC, accessed on 21 September 2021). UiT-TIL dataset and trained models: https://doi.org/10.18710/4YN9SZ; Open-access web server: https://hdl-histology-ne.azurewebsites.net/.

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
