# Peer review of "A Pragmatic Machine Learning Approach to Quantify Tumor-Infiltrating Lymphocytes in Whole Slide Images"

_cancers, 2022, doi:10.3390/cancers14122974_

Round 1

Reviewer 1 Report

This study proposed a pragmatic machine learning approach to quantify TILs in whole slide images, a hot and exciting topic. The proposed method provides one solution for quantifying TILs in WSIs and even further the deployment of the ML models in a cloud to potentially support the TILs evaluation in the clinical setting. My concerns are listed below:

1. The presentation of the manuscript needs substantial improvements. a. Figure 3 is missing; b. The abbreviations of Table 1 are suggested to be put on the same page; c. In the results part, "Manual validation of classification results," too many numbers in one paragraph make the reading experience not quite good, suggest reorganizing the presentation of results; d. some apparent typos, e.g., "iindicate" in the first sentence in the abstract. A careful re-editing is strongly suggested.

2. To evaluate the number of inflammatory/TILs for each patient, the authors propose to use the median number of all patches in the UiT-TILs. This is quite a reasonable idea since it can remove those the lower/higher extreme scores. But this simplification also discards substantial information within one whole slide after enormous computing. How to better combine the number of TILs from all patches could be one potential future study? Also, if focusing on the median value,  performing some patch samplings might be able to reduce the time costs and facilitate employment. 

3. The critical challenge for the TILs evaluation is to obtain accurate nuclei classification performance for the lymphocytes. The reimplemented HoVer-Net and additional augmentation all suggest the state-of-the-art performance of HoVer-Net as of now. It's reasonable to take these results to the downstream clinical setting. While how to further boost the cell (including but not limited to the tumor, immune, stroma, and more), classification would require continuously efforts. 

Author Response

We thank the reviewer for the constructive comments contributing to improve our manuscript. The specific points made are addressed below. Changes to manuscript has been made where it was deemed necessary.

1) The presentation of the manuscript needs substantial improvements. a. Figure 3 is missing; b. The abbreviations of Table 1 are suggested to be put on the same page; c. In the results part, "Manual validation of classification results," too many numbers in one paragraph make the reading experience not quite good, suggest reorganizing the presentation of results; d. some apparent typos, e.g., "iindicate" in the first sentence in the abstract. A careful re-editing is strongly suggested.

A1a: Figure 3 is now added to the manuscript
A1b: We agree and encourage Cancers type setters to make this adjustment in the final version of the manuscript.
A1c: The numbers are now omitted from the paragraph. In case of specific interest the reader is referred to the supplementary tables S5 and S6
A1d: Amendments including a comprehensive spell check have been made throughout the document

2. To evaluate the number of inflammatory/TILs for each patient, the authors propose to use the median number of all patches in the UiT-TILs. This is quite a reasonable idea since it can remove those the lower/higher extreme scores. But this simplification also discards substantial information within one whole slide after enormous computing. How to better combine the number of TILs from all patches could be one potential future study? Also, if focusing on the median value,  performing some patch samplings might be able to reduce the time costs and facilitate employment. 

A2: We thank the reviewer for this insightful comment. We are indeed pursuing this path of investigation in an expansion of our current work. A sentence proposing this future work has been incorporated into the "Future work" section of the manuscript

3. The critical challenge for the TILs evaluation is to obtain accurate nuclei classification performance for the lymphocytes. The reimplemented HoVer-Net and additional augmentation all suggest the state-of-the-art performance of HoVer-Net as of now. It's reasonable to take these results to the downstream clinical setting. While how to further boost the cell (including but not limited to the tumor, immune, stroma, and more), classification would require continuously efforts. 

A3: We agree with the reviewer's comment. The main advantage of HoVer-Net is segmentation. Since the model (per-pixel) really classifies pixels, it does not consider context including the class of neighboring cells and surrounding tissue structure. These and other factors should probably be incorporated in models to further improve classification performance. A sentence addressing these issues has been added to the "Future works" section.

Reviewer 2 Report

Improve resolution of images - make sure you export figures in 300dpi at least

You can find the rest (few) comments in the pdf i attached

Congratulations for the work. enjoyed reading it!

Author Response

We thank the reviewer for the positive reception of our work. Most of the proposed changes have been made throughout the manuscript. Specifically:

Figure 1: red and green is now changed to yellow and blue

Figure 3: is now added to the manuscript

Figure 4: is corrected so that legends do not overlap

Figure 5: We agree that blobs on a black background may increase interpretability for classification. However, it would be more difficult to assess differences in segmentation quality. Based on this we choose not to change this figure. 

Most of the numbers are now omitted from the paragraph "Manual validation of classifications results". In case of specific interest the reader is referred to the supplementary tables S5 and S6.

Typos and occasional grammatical errors have been amended throughout the manuscript.